# Neurological Manifestations of Neuropathy and Ataxia in Celiac Disease: A Systematic Review

**DOI:** 10.3390/nu11020380

**Published:** 2019-02-12

**Authors:** Elizabeth S. Mearns, Aliki Taylor, Kelly J. Thomas Craig, Stefanie Puglielli, Allie B. Cichewicz, Daniel A. Leffler, David S. Sanders, Benjamin Lebwohl, Marios Hadjivassiliou

**Affiliations:** 1IBM Watson Health, Cambridge, MA 02142, USA; elizabethmearns@gmail.com (E.S.M.); stefanie.puglielli@gmail.com (S.P.); allie.cichewicz@gmail.com (A.B.C.); 2Takeda Development Centre Europe Ltd., London WC2B 4AE, UK; aliki28@me.com; 3Takeda Pharmaceuticals International Co, Cambridge, MA 02139, USA; daniel.leffler@takeda.com; 4Royal Hallamshire Hospital and University of Sheffield, Sheffield S10 2RX, UK; david.sanders@sth.nhs.uk (D.S.S.); m.hadjivassiliou@sheffield.ac.uk (M.H.); 5Department of Medicine, Celiac Disease Center, Columbia University Medical Center, New York, NY 10032, USA; bl114@cumc.columbia.edu

**Keywords:** celiac disease, gluten neuropathy, gluten ataxia, prevalence, incidence, gluten-free diet

## Abstract

Celiac disease (CD) is an immune-mediated gastrointestinal disorder driven by innate and adaptive immune responses to gluten. Patients with CD are at an increased risk of several neurological manifestations, frequently peripheral neuropathy and gluten ataxia. A systematic literature review of the most commonly reported neurological manifestations (neuropathy and ataxia) associated with CD was performed. MEDLINE, Embase, the Cochrane Library, and conference proceedings were systematically searched from January 2007 through September 2018. Included studies evaluated patients with CD with at least one neurological manifestation of interest and reported prevalence, and/or incidence, and/or clinical outcomes. Sixteen studies were included describing the risk of gluten neuropathy and/or gluten ataxia in patients with CD. Gluten neuropathy was a neurological manifestation in CD (up to 39%) in 13 studies. Nine studies reported a lower risk and/or prevalence of gluten ataxia with a range of 0%–6%. Adherence to a gluten-free diet appeared to improve symptoms of both neuropathy and ataxia. The prevalence of gluten neuropathy and gluten ataxia in patients with CD varied in reported studies, but the increased risk supports the need for physicians to consider CD in patients with ataxia and neurological manifestations of unknown etiology.

## 1. Introduction

Celiac disease (CD) is a chronic, immune-mediated enteropathy in which dietary gluten triggers an inflammatory reaction of the small intestine in genetically predisposed individuals [1,2,3]. The clinical presentation of the disease varies broadly and may include an array of intestinal symptoms and extra-intestinal manifestations, such as iron-deficiency anemia, osteoporosis, dermatitis herpetiformis, and neurologic disorders [4]. 

Over the last several decades, the clinical presentation of CD has changed [5] with the proportion of patients presenting with classical CD symptoms decreasing and a corresponding increase in the frequency of extra-intestinal symptoms in children and adults [5,6,7]. This increasing proportion of extra-intestinal symptoms at presentation can result in lengthened diagnostic delay [8]. Active case-finding to facilitate prompt detection of CD and life-long adherence to a strict gluten-free diet (GFD) among patients with confirmed CD is recommended to reduce symptoms and the likelihood of disease of potentially serious manifestations [1]. 

Manifestations of CD can include a broad spectrum of musculoskeletal, neurological, cardiovascular, and autoimmune disorders. Most notably, peripheral neuropathies and gluten ataxia are frequent neurological manifestations of CD [9,10]. Many patients who present with neurological manifestations of CD have no gastrointestinal symptoms [11]. Peripheral neuropathy in patients with CD presents with tingling, pain, and numbness from nerve damage, initially in the hands and feet. Otherwise known as gluten neuropathy, it is defined as apparently sporadic idiopathic neuropathy in the absence of an alternative etiology and in the presence of serological evidence of gluten sensitivity. It is a slowly progressive disease with a mean age at onset of 55 years. Only one-third of patients have evidence of enteropathy on biopsy, but the presence or absence of enteropathy does not predetermine the effect of a GFD [12].

CD patients with ataxia often present with difficulty with arm and leg control, gait instability, poor coordination, loss of fine motor skills such as writing, problems with talking, and visual issues. Gluten ataxia usually has an insidious onset with a mean age at onset of 53 years [11]. Patients with gluten ataxia can show signs of cerebellar atrophy which can be irreversible and difficult to treat. Other neurological symptoms include encephalopathy, myopathy, myelopathy, ataxia with myoclonus, and chorea [9,10]. Gluten ataxia was first defined in 1996 as apparently idiopathic sporadic ataxia in patients with positive anti-gliadin antibodies (AGA). CD patients with gluten ataxia also often have oligoclonal bands in their cerebrospinal fluid, evidence of perivascular inflammation in the cerebellum, and anti-Purkinje cell antibodies [13].

Although these neurological manifestations of CD have been described over the last 30 years in the literature, there are still diagnostic delays often resulting in permanent neurological disability. Such delays are attributed to “controversies” arising from some variation in reported prevalence and poor understanding of the use of appropriate serological testing [14,15,16]. To examine the recent evidence, a systematic literature review was conducted to evaluate the prevalence and outcomes of the two most commonly reported neurological manifestations of CD: gluten neuropathy and gluten ataxia.

## 2. Materials and Methods

A systematic review of literature indexed in MEDLINE (via PubMed), Embase, and the Cochrane Library from January 2007 to August 2018 was performed in accordance with Preferred Reporting Items for Systematic Reviews and Meta-Analyses (PRISMA) guidelines [17]. The search strategy conducted in MEDLINE (via PubMed) is provided in Table 1. Manual backwards citation tracking of references from included studies and systematic review articles was performed to identify additional relevant studies. Searches were also performed in proceedings of the past three meetings (2015–2018) of the following conferences: Digestive Disease Week, American College of Gastroenterology, and United European Gastroenterology Week.

To be included, studies (primary studies or systematic reviews with or without meta-analyses) had to be conducted in patients with CD, published from 2007 or later (or last three meetings for conference abstracts) in English, and report the incidence, prevalence, and/or clinical outcomes of ataxia and/or neuropathy. Neuropathy, which is often used synonymously with peripheral neuropathy, is classified according to the type of damage to the nerve. In this systematic review, the terms “neuropathy” and “peripheral neuropathy” are stated as the authors have used them in their studies, recognizing that “neuropathy” may include a wider range of symptoms than peripheral neuropathy, which would represent a large proportion of neuropathy cases overall. 

A single investigator screened titles and abstracts to determine if the citation met inclusion criteria, with validation by a second reviewer required for exclusion. Two investigators independently reviewed all potentially relevant full-text citations, with discrepancies resolved by a third reviewer. Screening, data extraction, and validation were performed using DistillerSR (Evidence Partners Inc., Kanata, Ottawa, Canada). One investigator abstracted all data using a standardized tool, and a second reviewer verified entries. 

Two independent investigators assessed the quality of included studies using the Oxford Levels of Evidence Instrument [18]. Reviewers used the “Differential diagnosis/symptom prevalence study” section to assess the overall grade of the evidence. Details regarding the categorization of the study designs are available in Table 2. 

## 3. Results

The searches identified 441 citations, 299 conference abstracts, and 1 additional citation identified through backwards citation tracking. After removal of duplicates and screening of titles and abstracts screening, 45 were eligible for full-text review. A total of 16 studies met all eligibility criteria and were included in the systematic review (Figure 1, Table 3) [10,19,20,21,22,23,24,25,26,27,28,29,30,31,32,33]. Nine studies on gluten ataxia [10,20,21,22,23,26,28,31,32] and 13 articles on gluten neuropathy were included [10,19,20,22,24,25,26,27,28,29,30,32,33]. 

The PRISMA flow diagram depicts the flow of information through the different phases of this systematic review. It maps out the number of records identified, included and excluded, and the reasons for exclusions. 

Of the studies included, 50% (eight out of 16) were full-text, prospective analyses that reported global prevalence or incidence rates of gluten neuropathy and gluten ataxia [19,20,21,22,23,24,28,29]. See Box 1 for the definitions of CD, gluten ataxia, and gluten neuropathy. Most studies were performed in Europe (9; Germany, Italy, Romania, Sweden, and United Kingdom (UK)), while four were from the United States (US), two from Turkey, and one multinational study. Findings reported on adults (5), children (5), and both children and adults (6). Clinical outcomes of CD manifestations were reported in 50% (8 out of 16) of the included studies, while the remainder only addressed epidemiology. 

Box 1Definitions of gluten-sensitivity spectrum disorders used in this study.*celiac disease* – autoimmune disorder whereby gluten ingestion damages the portion of the small intestine responsible for nutrient absorption; also referred to as gluten-sensitive enteropathy. *gluten ataxia* – autoimmune disorder whereby gluten ingestion damages the cerebellum, which controls gait and muscle coordination, and fine control of voluntary movements is compromised. *gluten neuropathy* – autoimmune disorder whereby gluten ingestion damages the nerves of the peripheral nervous system, which disrupts communication from the brain and spinal cord to the rest of the body.

### 3.1. Gluten Neuropathy

Thirteen articles reported gluten neuropathy as a manifestation of CD [10,19,20,22,24,25,26,27,28,29,30,32,33]. Estimates of the prevalence of neuropathy in these patients ranged from 0% to 39%, with an increased prevalence/risk in older and female patients. In retrospective and prospective studies of patients with CD in the US and Europe, prevalence of neuropathy ranged from 4% to 23% of adults [20,25,27], 0% to 7% of children [22,24,25,28,32], and 0.7% to 39% of combined/unspecified populations [20,29,30,32]. While these ranges appear to overlap, a few studies directly compared the prevalence and risk of neuropathy by age and indicated that neuropathy occurs more frequently in older populations [27]. In a retrospective US study of adults (*n* = 171) and children (*n* = 157) with CD, gluten neuropathy was reported in 23% of adults with a follow-up period of >24 months between 2002 and 2014; however, no cases were reported in children [25]. Another retrospective US study found that significantly more elderly patients aged ≥65 years (11%) had gluten neuropathy compared with younger patients aged 18–30 years (4%; *p* = 0.023) [27]. Similar to young adults, gluten neuropathy was identified in 3 to 4.5% of children with CD in two studies [28,32]. Another questionnaire-based US study found that the risk of gluten neuropathy rose significantly with every ten-year increase in age (OR, 1.13; 95% CI, 1.04–1.23; *p* = 0.006). This study also reported a higher risk of gluten neuropathy in females (OR, 1.71; 95% CI, 1.25–2.33; *p* = 0.001) [29].

Gluten neuropathy may account for approximately one-quarter of neurological manifestations in those with CD. In two studies (one retrospective (*n* = 228) and one prospective (*n* = 72)) examining patients with CD and neurological conditions, gluten neuropathy accounted for 19% to 30% of neurological manifestations [10,20]. Patients with CD have a higher risk of gluten neuropathy and experience more severe neuropathic symptoms compared with non-CD controls (*p* < 0.01) [29]. In three studies (two retrospective and one questionnaire-based) from the US and Sweden, patients with CD had a significantly higher (2.3–5.6 times) risk of peripheral neuropathy compared with control populations [26,29,30]. The risk of polyneuropathy appears highest (4.4–5.6 times) during the first year of follow-up after CD diagnosis [26,30], compared with overall risk, or risk excluding the first year of follow-up (2.3–3.4 times) [26,30]. The risk estimate for neuropathy was only marginally affected after adjustment for education, socioeconomic status, type 1 diabetes mellitus (T1DM), type 2 diabetes mellitus (T2DM), thyroid disease, rheumatologic diseases, pernicious anemia, vitamin deficiencies, and alcoholic disorders (Hazard Ratio (HR), 2.3; 95% Confidence Interval (CI), 1.9–2.7) [30]. Notably, two of these studies adjusted their design to control for the rate of T1DM, as peripheral neuropathy is a long-term manifestation of T1DM [26,30]. However, Thawani et al. (2017) observed there was no significant increased risk of neuropathy for biopsy-confirmed CD patients with T1DM after examining neuropathy incidence in the first five years of CD diagnosis when compared to patients with T1DM only [33]. 

Symptoms from gluten neuropathy improve when patients with CD follow a GFD, although the diet may not prevent its development, and longer adherence to a GFD may not completely reverse neuropathy. One retrospective US study found that among patients who developed gluten neuropathy (*n* = 39), there was a significant improvement on a GFD (*p* < 0.05) [25]. Two prospective Italian studies also reported that in patients with gluten neuropathy, dietary adherence led to improvement in neuropathy and non-adherence led to worsening [20,28]. However, it should be noted that only one to two patients developed neuropathy in each of these Italian studies. While a GFD may improve symptoms of gluten neuropathy, one questionnaire-based US study found that duration of the diet (<5 vs. 5–9 vs. ≥10 years) did not significantly change the proportion of patients who developed the manifestation [29]. Similar proportions of patients developed neuropathy regardless of whether patients were reported to be following a GFD [10,22,25]. In the studies that did document GFD status, the extent of GFD adherence was not reported, limiting assessment of the relationship between neuropathy and degree of gluten exposure.

The severity of gluten neuropathy is variable. With a follow-up period of >20 years, one retrospective British study found that patients with CD on a GFD who developed gluten neuropathy, severity was mild (confined to the legs) in 27%, moderate (involvement of arms but sparing radial nerve) in 40%, and severe (involvement of radial nerve) in 33% [10]. A questionnaire-based US study suggested that the severity of neuropathy is not associated with duration on the GFD [29].

### 3.2. Gluten Ataxia

Upon physical examination for neurological deficits in patients with CD, estimates of the prevalence of gluten ataxia varied from 0% to 6% [20,21,22,23,28,32]. However, in studies among CD patients with neurological manifestations, gluten ataxia was reported in 19% to 41% of patients [10,23]. While studies tended to use similar definitions of ataxia, prevalence estimates varied. Six of the ten included studies used standard neurological exams with combinations of either magnetic resonance imaging (MRI) or magnetic resonance spectroscopy (MRS), or computed tomography (CT) to confirm the diagnosis of ataxia by examination of the vermis, eliminating other potential common causes of ataxia such as thyroid dysfunction, vitamin E deficiency, toxicity, and genetic forms of ataxia (spinocerebellar and Friedrich’s) [20,21,22,23,28,31]. 

Of the prospective European studies that used diagnostic CT or MRI/MRS, gluten ataxia was diagnosed in two studies [21,23]. One study of adults (*n* = 72) [21] and one of children (*n* = 48) [23] each reported a prevalence of 6% in patients with CD. The study of 48 children attributed the prevalence of gluten ataxia and the presence of the comorbidities of mental retardation and developmental delays to nutritional deficiencies and toxic effects of severe malnutrition [23]. The other three studies utilizing CT or MRI to define ataxia, one in adults (*n* = 71) [20] and two in children (*n* = 27 and *n* = 835) [22,28], reported that no patients (0%) developed ataxia. 

Two included retrospective studies did not report a prevalence of gluten ataxia [10,26]. One study used International Classification of Diseases (ICD, 7–10) codes to identify the symptom of ataxia (excluding trauma or toxicity as main diagnoses) or hereditary ataxia to determine the risk of ataxia in patients with CD [26]. The remaining study had less transparency in the diagnosis of ataxia as the diagnostic criteria were not described, where authors reported that a standard neurological assessment was performed and only reported on the severity of ataxia [10]. 

One British study suggested that most cases (69%) of gluten ataxia in patients with CD are mild, and patients could walk without assistance [10]. Of the remaining ataxia cases, 17% were moderate (requiring walking aids/support), and 14% were severe (needing a wheelchair). All patients were reported to be following a GFD [10].

In the nine included studies [10,20,21,22,23,26,28,31,32], gluten ataxia accounted for up to half of all neurological manifestations observed in people with CD. Definitive conclusions cannot be made regarding age-related differences in CD-associated ataxia from included studies, but available data suggest that gluten ataxia accounts for a smaller proportion of neurological manifestations in children with CD compared with adults. 

The risk of gluten ataxia appears to vary over time after CD diagnosis. A retrospective population-based registry study from Sweden evaluated the risk of gluten ataxia in patients with a hospital-based diagnosis of CD (*n* = 14,371), and found a greater risk of ataxia compared with controls without CD when patients were followed during the first year after discharge (HR, 2.6; 95% CI, 1.0–6.5; *p* = 0.042) [26]. However, if the first year of follow-up was excluded, the higher risk of ataxia was no longer statistically significant (HR, 1.9; 95% CI, 0.6–6.2; *p* > 0.05) based upon 14,371 patients with CD and 70,155 reference individuals [26]. 

The observed effect of GFD on ataxia may be dependent upon the methodological tests to monitor adherence to a GFD and the metrics utilized to assess neurological improvement. A quantitative assessment of the effect of GFD on gluten ataxia was provided by cerebellar MRS in Hadjivassiliou et al. (2017) [31]. In this study, CD patients with gluten ataxia (*n* = 117) were reviewed for response to GFD: 63 were on strict GFD with the elimination of AGAs, 35 were on GFD but still positive for AGAs, and 19 patients were not on a GFD. GFD adherence was monitored by serological assessments. On MRS, there was a significant improvement in the cerebellum in 62 out of 63 (98%) patients on a strict GFD, in nine of 35 (26%) patients on GFD with positive AGAs, but in only one of 19 (5%) patients not on GFD. Notably, the presence of enteropathy (CD), usually required for the diagnosis of CD, in addition to positive serology, was not found to be a prerequisite for improvement in the cerebellum. The authors concluded that patients with positive serology results and negative duodenal biopsy should still be treated with strict GFD and noted that improved cerebellar function with GFD adherence was associated with clinical improvement [31]. In contrast, a prospective Romanian study in 48 children reported that none of the patients with gluten ataxia had improved symptoms while on a GFD [23]. However, Diaconu et al. (2014) did not state how GFD adherence was monitored and ataxia assessments were self-reported by the parents of the children affected [23].

### 3.3. Quality Assessment

Based on Oxford Levels of Evidence, the evidence in this review has an overall grade of B. Only one study provided Level 1b evidence [19]. Seven studies [10,25,26,27,30,32,33] were retrospective cohort studies, which represented Level 2b evidence. One study was a prospective case series, representing level 4 evidence [31]. The remaining seven studies [20,21,22,23,24,28,29] were cross-sectional studies, which we have categorized as Level 2c. The levels of evidence for individual studies are shown in Table 4.

## 4. Discussion

This systematic review demonstrates that gluten neuropathy was reported more often than gluten ataxia (81.25% of included studies reported neuropathy), although the prevalence of gluten neuropathy varied widely (0%–39%). Both ataxia and neuropathy were more prevalent in patients with CD compared with controls. Symptoms of neuropathy were most commonly categorized as moderate, affecting extremities. Prevalence of gluten ataxia in patients with diagnosed CD varied from 0–6%; symptoms were often described as mild, in which patients were still able to walk, although in some cases could be very severe and persistent. The variations in prevalence rates across studies of both gluten ataxia and gluten neuropathy may be related to study design and inclusion criteria, retrospective nature of data collection, quality of assessment of adherence to a GFD, clinical assessment of neurological symptoms, and the age of the populations included. 

The prevalence of idiopathic neuropathy in the general population is low but the risk is increased in CD. A literature review of 28 studies reported the prevalence of neuropathy in the general middle-aged and elderly population between 0.1% and 3.3% [34]. Increased neuropathy prevalence was reported in a US study published in 2003 using retrospective data from 400 patients with neuropathy, whereby neuropathy rates for CD were between 2.5% and 8% (compared to 1% in the healthy population) [35]. In a large Swedish population-based study that examined the risk of neurological disease, polyneuropathy was found to be significantly associated with CD (odds ratio 5.4; 95% CI 3.6–8.2) [36]. In further support of this, an age- and sex-matched control study, identified in this review, comparing patients with CD to controls found that CD was associated with a 2.5-fold increased risk of later neuropathy [30]. The highest risk for gluten neuropathy was just after diagnosis of CD, but there was also a consistent excess risk of neuropathy beyond five years after a diagnosis of CD. Two other included studies compared patients with CD of different ages and found that younger patients were less likely to experience neuropathy [25,27]. However, these studies examined established patients with CD and their findings may be an underestimation of risk of neuropathy in young patients. The presentation of atypical symptoms, such as neurological complications, at time of CD diagnosis in children, reported neuropathy prevalence of 10.5% in this small study population [32]. 

Similar to trends for neuropathy, the prevalence of ataxia in the general population is very low, but this risk is increased in patients with CD. A UK based population-based study estimated the prevalence of late-onset cerebellar ataxia as 0.01% in the general population [37]. Three studies identified in this review reported no cases (0%) of ataxia in both adults and children [20,22,28]; however, estimates of ataxia prevalence ranged from 0-6% across all ages [21,23,32]. In studies that determined ataxia prevalence in children, neurological manifestations were the initial symptoms of CD in 25%–33.33% of patients, and ataxia accounted for 5.26%–18.8% of those cases. [23,32]. The risk of ataxia in those with CD was estimated to be 1.9- to 2.6-fold compared with controls during the first year after diagnosis [26]. 

Although the prevalence of ataxia in CD is thought to be low, it may be underestimated. A recent UK study of 500 patients diagnosed with progressive ataxia and evaluated over a period of 13 years, found that 101 of 215 (47%) patients with idiopathic sporadic ataxia had serological evidence of gluten reactivity [38]. A study of 1500 patients with cerebellar ataxia referred to the Sheffield Ataxia Centre, UK assessed over 20 years found that 20% had a family history of ataxia, and the remaining 80% had sporadic ataxia. Of sporadic ataxias, gluten ataxia was the most common cause (25%); followed by genetic causes (13%), alcohol excess (12%), and a cerebellar variant of multiple system atrophy (11%) [39]. In a review of gluten sensitivity by Hadjivassiliou et al. (2010) [11], many studies reported that a high proportion of patients with sporadic ataxias (12%–47%) tested positive for AGA compared with 2%–12% of healthy controls [11,38,39,40,41,42,43,44,45,46,47,48]. These studies suggest that even though ataxia is rare, gluten ataxia is a common subtype of sporadic ataxia. 

Adherence to a strict GFD can result in clinical improvement in both gluten neuropathy and gluten ataxia. Publications which met criteria for inclusion in this review unanimously support a beneficial effect of the GFD on neuropathy, however, a benefit in ataxia is less clear. Some studies report that ataxia persists in patients on a GFD, while others demonstrated improvement on GFD [10,21,23,31]. This heterogeneity is most likely due to differences in study design, including the assessment of GFD adherence and ataxia symptoms. Severity of ataxia can be assessed with a variety of instruments including self-report and clinician determination using scales for the assessment and rating of ataxia (e.g., Brief Ataxia Rating Scale (BARS), Scale for the Assessment and Rating of Ataxia (SARA), International Cooperative Ataxia Rating Scale (ICARS), modified ICARS (MICARS)), and imaging studies (e.g., MRS, MRI, EEG). Objective quantitation of motor deficits in ataxia is fundamental for measurement of clinical severity but was not commonly reported in studies examining the association between improvements of ataxia and GFD adherence. One study by Hadjivassiliou et al. (2017) utilized a quantitative methodology via MRS to monitor ataxia severity by cerebellar atrophy and assessed GFD adherence with AGA testing [31]. This study demonstrated a beneficial effect of strict GFD adherence on ataxia and benefits were seen in all AGA positive individuals, regardless of baseline enteropathy [31]. 

It is important to clarify the differences between CD and gluten sensitivity in the context of gluten ataxia and gluten neuropathy. This systematic review primarily concentrated on patients with CD and these two common neurological manifestations. These manifestations, however, may exist in the presence of AGA alone (gluten sensitivity) without evidence of enteropathy (CD), and such patients benefit equally from GFD. Indeed Hadjivassiliou et al. (2016) demonstrated there are no distinguishing features (e.g., type of neurological manifestation, severity, and response to GFD) between those patients with neurological manifestations and CD and those with just positive AGA (no enteropathy) [10]. Despite this, the majority of immunological laboratories have abandoned the use of native AGA assays due to poor specificity in diagnosing CD. Estimation of specificity, however, is based on the presence of a gold standard, in this case, the presence of enteropathy. Given that sensitivity to gluten exists in the absence of enteropathy, then AGA remains probably the only serological marker in diagnosing the whole spectrum of extraintestinal manifestations. Another important consideration when using AGA is the serological cut-off for positive AGA. Such assays are calibrated using serology from patients with CD as the gold standard, and consequently, the serological cut-off tends to be high. It has recently been shown that by recalibrating the serological cut-off of a commercially available AGA assay based on the ability to diagnose GA, the sensitivity of AGA in diagnosing CD became 100% [49].

There were a small number of studies identified that did not meet our inclusion criteria but described the association between gluten neuropathy and enteropathy, and the effects of strict GFD on gluten neuropathy. Of note, a study published by Hadjivassiliou et al. (2006) reported that of 100 patients with clinical immunological characteristics of gluten neuropathy, 29% of patients had evidence of enteropathy [50]. A prospective study published in 2006, followed 35 patients with gluten neuropathy, 25 of which were assigned to strict adherence to a GFD with the remaining ten patients as controls. Strict GFD adherence was defined by the elimination of AGA after one year. When asked, 16/25 patients on the GFD said their neuropathy was better compared to 0/10 in the control group. Eight out of ten patients in the control group stated that their neuropathy was worse [12]. Gluten neuropathy can be associated with significant chronic pain and negatively impact mental health. A recent study assessed neuropathic pain in 60 patients with gluten neuropathy. Neuropathic pain was present in 33 patients and painless neuropathy was more common in patients on a strict GFD (55.6% versus 21.2%, *p* = 0.006). Patients with painful gluten neuropathy presented with significantly worse mental health status [12]. Multivariate analysis showed that, after adjusting for age, gender and mental health index-5, strict GFD was associated with an 89% reduction in risk of peripheral neuropathic pain (*p* = 0.006) [51].

Gluten ataxia and neuropathy were selected for this review because they are the most common neurological manifestations in CD. However, there are other neurological manifestations not assessed (a systematic review of movement disorders related to gluten sensitivity by Vinagre-Aragon et al. (2017) is available [52] for reference). A prospective study reported that up to 22% of patients with CD (*n* = 71) developed some form of neurologic or psychiatric dysfunction (headache, depression, entrapment syndromes, peripheral neuropathy, and epilepsy) [20]. In a British study published in 1998, 57% of patients with neurological dysfunction of unknown cause had serological evidence of gluten sensitivity, compared with 12% of healthy blood donors [53]. Neurological manifestations can have a significant impact on patients’ quality of life, and a greater understanding of these complications is needed. 

There are several limitations to this systematic review. Both clinical and methodological heterogeneity among reviewed studies limited comparisons of the data. Across all studies included, it is not possible to determine whether factors such as the timing of diagnosis, presentation of CD, or differences diagnostic techniques, affect rates of ataxia and peripheral neuropathy. Lastly, there is potential for publication bias and missed eligible articles in any literature review. However, this risk is assumed to be minimal due to strict adherence to standards for systematic search methodology. 

## 5. Conclusions

In conclusion, this systematic review provides important evidence on the substantially increased risk of gluten ataxia and gluten neuropathy in patients with CD, although estimates across studies vary. These results indicate that adherence to a GFD appears to improve symptoms of both neuropathy and ataxia. The scarcity of data from this global search highlights the need for additional well-designed studies to improve the understanding of neurological manifestations in patients with CD. Given that these results suggest an increased risk of ataxia and neuropathy among patients with CD, clinicians should evaluate for gluten sensitivity in patients with ataxia and neuropathy of unknown origin. 

## Figures and Tables

**Figure 1 nutrients-11-00380-f001:**
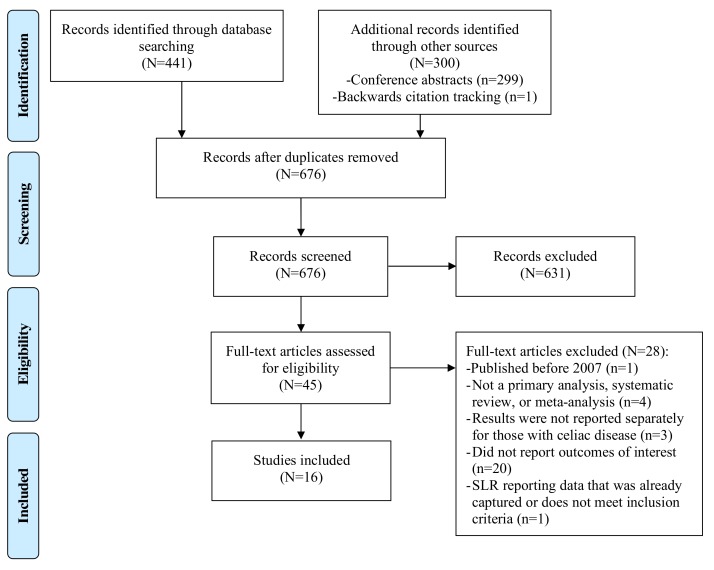
Flow Diagram Showing the Results of the Literature Search.

**Table 1 nutrients-11-00380-t001:** MEDLINE (via PubMed.com) Search Strategy.

Search No.	Search Terms	Search Results (28 August 2018)
1	celiac*[tiab] OR coeliac*[tiab] OR celiac disease[MeSH]	31,137
2	((coeliac OR celiac) AND (trunk* OR axis OR node*)) OR “coeliac artery” OR “celiac artery”	7348
3	#1 NOT #2	25,521
4	(cerebellar ataxia[MeSH] OR “cerebellar ataxia”[tiab] OR ((cerebellum* OR cerebellar) AND ataxi*) OR “gluten ataxia” OR “gluten-sensitive ataxia”)	17,238
5	neuropathy[tiab] OR neuropathies[tiab] OR neuropathic[tiab]	88,749
6	#3 AND #4	141
7	#3 AND #5	213
8	#6 OR #7	309
9	case reports[pt]	1,893,340
10	#8 NOT #9	238
11	mice OR mouse OR murine OR rodent*	1,754,334
12	#10 NOT #11	227
13	review[pt] NOT (systematic OR Cochrane OR meta-analy*)	2,162,485
14	#12 NOT #13	160
15	#14; Filter: published 2007 or later	99
16	#15; Filter: abstract	96

Footnotes: *, wildcard search term; #, search number. Abbreviations: tiab, title/abstract; pt, publication type.

**Table 2 nutrients-11-00380-t002:** Oxford Levels of Evidence & Grades of Recommendation.

**Level**	**Differential Diagnosis/Symptom Prevalence Study**
1a	Systematic review (with homogeneity) of prospective cohort studies
1b	Prospective cohort study with good follow-up
1c	All or none case-series
2a	Systematic review (with homogeneity) of 2b and better studies
2b	Retrospective cohort study, or poor follow-up
2c	Ecological studies
3a	Systematic review (with homogeneity) of 3b and better studies
3b	Non-consecutive cohort study, or very limited population
4	Case-series or superseded reference standards
5	Expert opinion without explicit critical appraisal, or based on physiology, bench research or “first principles”
**Grade**	**Levels of Individual Studies**
A	Consistent level 1 studies
B	Consistent level 2 or 3 studies or extrapolations from level 1 studies
C	Level 4 studies or extrapolations from level 2 or 3 studies
D	Level 5 evidence or troublingly inconsistent or inconclusive studies of any level

Table adapted from the Oxford Centre for Evidence-Based Medicine [18].

**Table 3 nutrients-11-00380-t003:** Characteristics of Included Studies.

Author (year)	Study Design	Country	Population	Neurological Complication
Briani and Doria et al. (2008) [19]	Prospective, single-center, cross-sectional	Italy	Patients with CD	Neuropathy
Briani and Zara et al. (2008) [20]	Prospective, single-center, cross-sectional	Italy	Unselected, consecutive patients with CD treated at the University of Padova	Ataxia, neuropathy
Burk et al. (2013) [21]	Prospective, single-center, cross-sectional	Germany	Patients with CD on a GFD recruited from advertisements in the official journal of the German Celiac Society or personal contact	Ataxia
Cakir et al. (2007)[22]	Prospective, multi-center, cross-sectional	Turkey	Children with CD treated at the outpatient follow-up program of celiac patients in the pediatric gastroenterology and nutrition division of Ege University Hospital from 1998–2002	Ataxia, neuropathy
Diaconu et al. (2013) [23]	Prospective, single-center, cross-sectional	Romania	Children (2–18 years) diagnosed with CD from 2000–2010	Ataxia
Hadjivassiliou et al. (2016) [10]	Retrospective, single-center, cohort	UK	Patients with CD and neurological manifestations presenting to the Neuroscience Department at Royal Hallamshire Hospital from 1994–2014	Ataxia, neuropathy
Hadjivassiliou et al. (2017) [31]	Prospective, single-center, observational case series	UK	Patients diagnosed with gluten ataxia at the Sheffield Ataxia Centre	Ataxia
Isikay et al. (2015) [24]	Prospective, single-center, cross-sectional, case-control	Turkey	Asymptomatic children with CD diagnosed at a pediatric gastroenterology outpatient clinic from September 2012–August 2014	Ataxia, neuropathy
Jericho et al. (2017) [25]	Retrospective, single-center, chart review	US	Patients with CD registered at the University of Chicago Celiac Center clinic from January 2002–October 2014	Ataxia, neuropathy
Ludvigsson et al. (2007) [26]	Retrospective, multi-center, database	Sweden	Patients in the Swedish national inpatient register with a hospital-based discharge diagnosis of CD from 1964–2003	Ataxia, neuropathy
Mukherjee et al. (2010) [27]	Retrospective, single-center, database	US	Patients with CD from a prospectively generated database at a university-based referral center	Neuropathy
Ruggieri et al. (2008) [28]	Prospective, single-center, cross-sectional	Italy	Children with CD and neurological dysfunction evaluated at the gluten sensitivity clinic at the Department of Pediatrics at the University of Catania from January 1991–December 2004	Ataxia, neuropathy
Sangal et al. (2017) [32]	Retrospective, single-center, medical record review	Not reported	Children with and without gluten-related disorders between July 2013 and May 2016	Ataxia, neuropathy
Shen et al. (2012) [29]	Questionnaire-based, multi-center, cross-sectional, case-control	US	Patients with CD recruited from the Celiac Disease Center at Columbia University and support groups in New York and California	Neuropathy
Thawani et al. (2015) [30]	Retrospective, multi-center	Sweden	Patients with CD from one of Sweden’s pathology departments from June 1969–February 2008	Neuropathy
Thawani et al. (2017) [33]	Retrospective, nationwide registry	Sweden	Patients diagnosed with T1DM between 1964 and 2009, with and without CD (based on biopsies between 1969 and 2008) in the Swedish National Patient Register	Neuropathy

Abbreviations: CD, celiac disease; GFD, gluten-free diet; T1DM, type 1 diabetes mellitus; US, United States; UK, United Kingdom.

**Table 4 nutrients-11-00380-t004:** Quality Assessment of Included Studies.

Study Identifier	Oxford Level of Evidence
Briani and Doria et al. (2008) [19]	2c. Ecological study *
Briani and Zara et al. (2008) [20]	1b. Prospective cohort study
Burk et al. (2013) [21]	2c. Ecological study *
Cakir et al. (2007) [22]	2c. Ecological study *
Diaconu et al. (2013) [23]	2c. Ecological study *
Hadjivassiliou et al. (2016) [10]	2b. Retrospective cohort study
Hadjivassiliou et al. (2017) [31]	4. Case-series or superseded reference standards
Isikay et al. (2015) [24]	2c. Ecological study *
Jericho et al. (2017) [25]	2b. Retrospective cohort study
Ludvigsson et al. (2007) [26]	2b. Retrospective cohort study
Mukherjee et al. (2010) [27]	2b. Retrospective cohort study
Ruggieri et al. (2008) [28]	2c. Ecological study *
Sangal et al. (2017) [32]	2b. Retrospective cohort study
Shen et al. (2012) [29]	2c. Ecological study *
Thawani et al. (2015) [30]	2b. Retrospective cohort study
Thawani et al. (2017) [33]	2b. Retrospective cohort study

*, Note that this was a cross-sectional study, not an ecological study; there is no Oxford Level of Evidence for cross-sectional studies [18].

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
