# Peer review of "Neurological Manifestations of Neuropathy and Ataxia in Celiac Disease: A Systematic Review"

_nutrients, 2019, doi:10.3390/nu11020380_

Reviewer 1 Report

This is a very interesting and useful systematic review on two neurological complications in celiac disease : peripheral neuropathy and gluten ataxia. This systematic review is well designed , conducted, analysed, and presented.

Author Response

Thank you for the positive feedback. No questions or comments were posed by the reviewer to address in our revision.

Reviewer 2 Report

The manuscript is a thorough review about the existing literature considering the prevalence and clinical outcomes of gluten ataxia and neuropathy, which are the two most common neurological manifestations of celiac disease. The literature search has been made meticulously, and the article selection and quality assessment are well described in the manuscript. In general, the paragraphs are concise and include appropriate amount of information for the reader. The graphics are appropriate and supplement the text well. The reference list is comprehensive.  

The only thing I am a little bit concerned about is that the definition of gluten-related neuropathy and ataxia, and especially their relations to celiac disease in general, might confuse the reader. It is stated that gluten-related neuropathy and ataxia are idiopathic forms of the conditions with no other etiology and in the presence of serological evidence of gluten sensitivity. Majority of the referred studies have used anti-gliadin antibodies instead of anti-tTG or EmA. However, anti-gliadin antibodies are not part of the current serological diagnostics of celiac disease due to their insufficient sensitivity and specificity (e.g. Husby et al. 2012). Celiac disease researchers are familiar with the suggestion that these neurological manifestations somehow differ from celiac disease in general and may more often present without transglutaminase or endomysium antibody positivity. However, this might be less familiar among the readers working in clinical practice, as well as might be with the overlapping of terms such as celiac disease, gluten sensitivity and gluten-spectrum disorders (as defined e.g. in the Oslo definitions for coeliac disease and related terms). Consequently, I think that it might be a good idea to briefly discuss and clarify to the reader, why AGA:s are used in the diagnostics of gluten-related neurological manifestations but not in celiac disease, what is their relations with celiac disease, and what possible problems the differences in serology may bring to the diagnostics and estimated prevalence of these conditions (is ther for example some risk of wrong positive diagnosis, especially if the condition seems to progress during a GFD?).

Author Response

Point 1: The manuscript is a thorough review about the existing literature considering the prevalence and clinical outcomes of gluten ataxia and neuropathy, which are the two most common neurological manifestations of celiac disease. The literature search has been made meticulously, and the article selection and quality assessment are well described in the manuscript. In general, the paragraphs are concise and include appropriate amount of information for the reader. The graphics are appropriate and supplement the text well. The reference list is comprehensive.  

Thank you for the positive feedback.

Point 2: The only thing I am a little bit concerned about is that the definition of gluten-related neuropathy and ataxia, and especially their relations to celiac disease in general, might confuse the reader. It is stated that gluten-related neuropathy and ataxia are idiopathic forms of the conditions with no other etiology and in the presence of serological evidence of gluten sensitivity. Majority of the referred studies have used anti-gliadin antibodies instead of anti-tTG or EmA. However, anti-gliadin antibodies are not part of the current serological diagnostics of celiac disease due to their insufficient sensitivity and specificity (e.g. Husby et al. 2012). Celiac disease researchers are familiar with the suggestion that these neurological manifestations somehow differ from celiac disease in general and may more often present without transglutaminase or endomysium antibody positivity. However, this might be less familiar among the readers working in clinical practice, as well as might be with the overlapping of terms such as celiac disease, gluten sensitivity and gluten-spectrum disorders (as defined e.g. in the Oslo definitions for coeliac disease and related terms).

 Consequently, I think that it might be a good idea to briefly discuss and clarify to the reader, why AGA:s are used in the diagnostics of gluten-related neurological manifestations but not in celiac disease, what is their relations with celiac disease, and what possible problems the differences in serology may bring to the diagnostics and estimated prevalence of these conditions (is ther for example some risk of wrong positive diagnosis, especially if the condition seems to progress during a GFD?).

This was a helpful suggestion to provide clarification for the readers. The following information was added to the discussion section (lines 313-329):

            “It is important to clarify differences between CD and gluten sensitivity in the context of gluten ataxia and gluten neuropathy. This systematic review primarily concentrated on patients with CD and these two common neurological manifestations. These manifestations, however, may exist in the presence of AGA alone (gluten sensitivity) without evidence of enteropathy (CD), and such patients benefit equally from GFD. Indeed Hadjivassiliou et al. (2016) demonstrated there are no distinguishing features (e.g., type of neurological manifestation, severity, and response to GFD) between those patients with neurological manifestations and CD and those with just positive AGA (no enteropathy) [Hadjivassiliou et al., (2016) Am J Gastroenterology]. Despite this, the majority of immunological laboratories have abandoned the use of native AGA assays due to poor specificity in diagnosing CD. Estimation of specificity, however, is based on the presence of a gold standard, in this case the presence of enteropathy. Given that sensitivity to gluten exists in the absence of enteropathy, then AGA remain probably the only serological marker in diagnosing the whole spectrum of extraintestinal manifestations. Another important consideration when using AGA is the serological cut-off for positive AGA. Such assays are calibrated using serology from patients with CD as the gold standard, and consequently, the serological cut-off tends to be high. It has recently been shown that by recalibrating the serological cut-off of a commercially available AGA assay based on ability to diagnose GA, the sensitivity of AGA in diagnosing CD became 100% [New Ref added, Hadjivassiliou et al., (2018) Nutrients].”